## Research Article

**Key words**
Insulin secretory granule; integrative mesoscale modelling; pancreatic beta cell; structural biology

**Author for correspondence:**
David S. Goodsell,
E-mail: goodsell@scripps.edu

# Integrative structural modelling and visualisation of a cellular organelle

Ludovic Autin[1] ◉, Brett A. Barbaro[1] ◉, Andrew I. Jewett[1] ◉, Axel Ekman[2] ◉, Shruti Verma[1] ◉, Arthur J. Olson[1] ◉ and David S. Goodsell[1,3,4] ◉

[1]Department of Integrative Structural and Computational Biology, The Scripps Research Institute, La Jolla, CA 92037, USA; [2]Department of Anatomy, University of California, San Francisco, San Francisco, CA 94143, USA; [3]Research Collaboratory for Structural Bioinformatics Protein Data Bank, Rutgers, The State University of New Jersey, Piscataway, NJ 08854, USA and [4]Institute for Quantitative Biomedicine, Rutgers, The State University of New Jersey, Piscataway, NJ 08854, USA

## Abstract

Models of insulin secretory vesicles from pancreatic beta cells have been created using the cellPACK suite of tools to research, curate, construct and visualise the current state of knowledge. The model integrates experimental information from proteomics, structural biology, cryoelectron microscopy and X-ray tomography, and is used to generate models of mature and immature vesicles. A new method was developed to generate a confidence score that reconciles inconsistencies between three available proteomes using expert annotations of cellular localisation. The models are used to simulate soft X-ray tomograms, allowing quantification of features that are observed in experimental tomograms, and in turn, allowing interpretation of X-ray tomograms at the molecular level.

## Introduction

Structural modelling of entire cells is a new frontier actively being addressed by the structural biology community. Given the magnitude and complexity of this challenge, it is essential to take an integrative approach, bringing together data from multiple experimental modalities that address multiple levels of temporal and spatial scales. Already, this approach has generated detailed atomic models of complex enveloped viruses, entire bacterial cells and cellular organelles, as reviewed in Im *et al.* (2016), Feig and Sugita (2019), and Goodsell *et al.* (2020).

A recent perspective (Singla *et al.,* 2018) identified several characteristics that are needed for an effective integrative mesoscale structural model. The model needs to be complete over multiple levels of scale, from atomic details to overall ultrastructure. The model needs to couple a variety of representations in order to integrate different modalities of structural, biochemical, physiological and bioinformatics knowledge. Furthermore, as a key part of the process of integration, the uncertainty of each parameter defining the model needs to be quantified and made accessible in analysis and visualisation of the model. Finally, the model must capture aspects of the heterogeneity of the system, allowing it to be both descriptive and predictive.

As part of the Pancreatic Beta-Cell Consortium (PBCC; pbcconsortium.org), we are developing methods to generate mesoscale models of functional regions of the pancreatic beta cell based on diverse experimental data from the PBCC and the larger research community. As our first proof of concept, we have chosen to model one of the defining characteristics of this cell, the insulin secretory granule (ISG). This is a particularly amenable initial target, given the abundance of available information, its manageable size and complexity and its functional connection to disease states of the cell. With this report, we present an entire pipeline from data curation to model generation, and present potential applications that are facilitated by this quantitative approach to mesoscale cellular biology. We also show preliminary work to simulate soft X-ray tomograms of vesicles. Soft X-ray tomography is an attractive experimental technique for imaging whole cells, since the experiment is performed in 'near-native' conditions with no fixatives or freezing (McDermott *et al.,* 2009) and is being actively applied to pancreatic beta cells in the PBCC (White *et al.,* 2020; Loconte *et al.,* 2022). We are exploring the use of integrative modelling to provide molecule-level interpretation of features in these tomograms, which typically have a resolution of 50–60 nm.

### State of knowledge for the insulin secretory granule

Given its central role in the regulated delivery of insulin, there is abundant information available for ISG structure and function. We relied on several excellent reviews to provide general synthesis of current knowledge (Suckale and Solimena, 2010; Germanos *et al.,* 2021). In the generally accepted view, immature ISGs have a single membrane and are filled with proinsulin and other

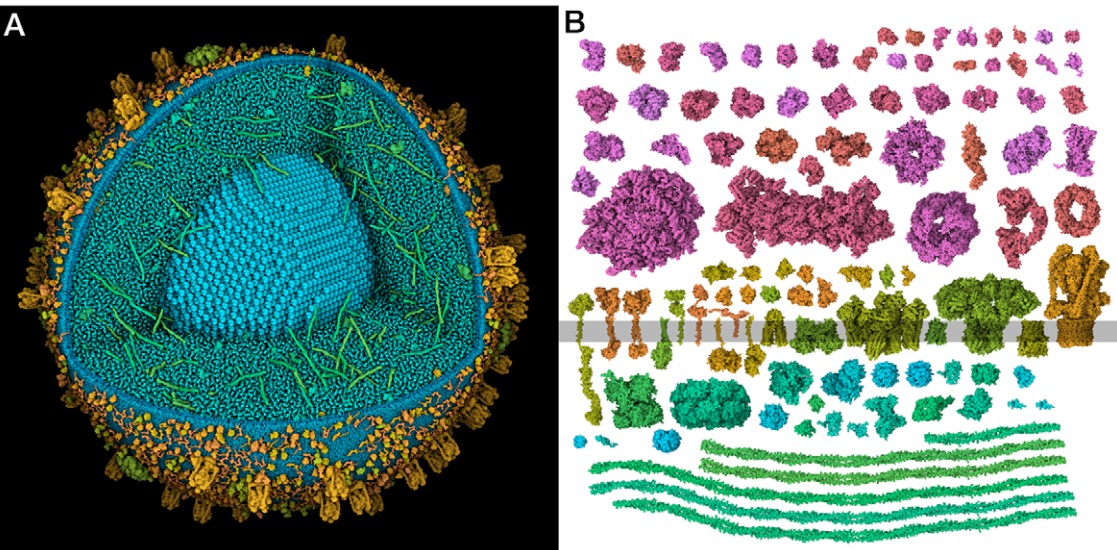

**Figure 1.** (*a*) Integrative 3D model of a mature insulin secretory granule. One quadrant is clipped away to show the insulin crystal (turquoise). The long coiled-coil proteins in green are granins, and the lumen is dominated by many copies of the small beta-peptide left over from the maturation of insulin. (*b*) Structural proteomes used to build the models. Cytoplasmic proteins are at the top in red and magenta, vesicle membrane-spanning and membrane-associated proteins are at the centre in orange and yellow-green and vesicle lumen proteins are at the bottom in blue-green.

proteins, which then mature under the action of several specific proteases. Insulin then crystallises into a single crystal that fills most of the interior. A complex interaction with the cytoskeleton mediates storage and delivery of ISGs to the cell surface, and fusion results in dissolution of the insulin crystal and release into the blood. In the current work, we are limiting our modelling effort to just the ISG membrane and interior, in immature and mature forms, and a generic cytoplasm.

However, there is still much uncertainty about basic information, such as the ISG proteome components, concentrations and interactions, as described in more detail below. Several proteomes are available that show discrepancies that must be reconciled. Even basic numbers, such as the number of insulin molecules per vesicle, show wide ranges, for example, from 200 thousand (Suckale and Solimena, 2010) to 1 million (Eliasson *et al.,* 2008). Much of the work presented in this paper is seeking to resolve this uncertainty to generate molecular models of the entire organelle that are consistent with the current state of knowledge (Fig. 1*a*).

In this work, we gathered, curated and integrated current information and used it to build idealised models of an ISG in cytoplasm. A manual literature search identified the 29 most widely reported ISG protein components. These 29 were then used to train a method for scoring proteins from three proteomes, identifying 14 additional proteins that have evidence for association with the ISG but were missed in the manual search. In addition, 13 proteins not included in the proteomes, but with experimental data supporting localisation in the ISG, were identified by a manual literature search. Structures for these 56 proteins were identified or modelled (Fig. 1*b*), and were then used to build 3D models of idealised immature and mature ISGs. Finally, X-ray tomograms were simulated from the models and used to classify features in experimental tomograms of pancreatic beta cells.

## Methods

### *Confidence scores for proteomics data*

Results from three proteomes were reconciled using a simple scoring function that combines the proteomics statistics and location

annotations into a confidence value. In brief, high scores are assigned to proteins found in multiple proteomes, that have annotations related to secretory vesicles, and that interact with proteins that also have annotations related to secretory vesicles. The set of 29 manually curated proteins (Supplementary Table 1) was assigned as true positives, and the remaining proteins from the three proteomes were assigned as trial negatives. Then, for each location annotation (loc = 'nucleus', 'cytoplasm' etc.), a location score [LocScore(loc)] was evaluated:

| | |
|---|---|
| $TPR(loc) = P(loc)/P_{tot}$ | True positive rate |
| $P(loc)$ | Number of true positives with a particular location annotation |
| $P_{tot}$ | Total number of true positives |
| $FPR(loc) = N(loc)/N_{tot}$ | False positive rate |
| $N(loc)$ | Number of false positives with a particular location annotation |
| $N_{tot}$ | Total number of negatives |
| $LocScore(loc) = [TPR(loc) - FPR(loc)]/[TPR(loc) + FPR(loc)]$ | |

A similar score, IntLocScore(intloc), was calculated for the location of proteins that have been annotated as interacting with the protein of interest. This is based on the assumption that interacting proteins will be found in the same compartment. The confidence of a protein is calculated by

$$confidence = N_{proteome}/3 + W_{loc} \times \sum LocScore(loc)$$
$$+ W_{intloc} \times \sum IntLocScore(intloc),$$

where the sum is performed over all of the location annotations for the particular protein and $N_{proteome}$ is the number of proteomes that include the protein.

A parameter sweep of the two weights was performed, yielding values of $W_{loc} = 12$ and $W_{intloc} = 1$ for the best Receiver Operating Characteristic (ROC) value over the entire dataset. The entire set of

proteins was then rescored based on these weights, and the scores were used for ranking the set. Given the small size of the set, cross-validation studies were not performed, so we expect that the resulting high scores of the true positives will be due to bias from use as the training set.

Location annotations were found by programmatically querying UniProt (https://www.uniprot.org) with gene names from each protein and rat taxid 10116 to get the UniProt entry with the highest annotation score, and extract location annotations listed under 'Keywords – Cellular component'. We then programmatically queried the UniProt, StringDB, Biogrid and Intact databases to get a list of proteins that interact with the protein being evaluated, and extracted location annotations for these in the same way. This process was repeated for human taxid 9096 and mouse taxid 10090. Annotations and fractions of positives and negatives for the training set are included in Supplementary Table 2.

### Protein structures and concentrations

To estimate relative concentrations within each proteomic study, 'Mascot Protein Score' values were used for proteome 1 (Brunner *et al.,* 2007), 'score' values for proteome 2 (Hickey *et al.,* 2009) and 'Intensity HiC12mC13' values for proteome 3 (Schvartz *et al.,* 2012). These values were divided by the length of the protein and then by the sum of all values to create a 'normalised spectral abundance factor' (NSAF) for each entry, which indicates roughly the molar fraction for each protein detected in the sample. Each protein's NSAF was then multiplied by its molecular weight and normalised again to determine its mass fraction. Proteins that were identified by manual curation but not observed in the proteomics studies were assigned arbitrarily low concentration values. Finally, the total mass fraction of non-insulin proteins was scaled to give a total mass fraction of 0.2, with insulin the crystal accounting for the remaining 0.8 mass fraction (Hutton, 1989).

Absolute copy numbers for proteins were calculated for a generic ISG with a crystal diameter of 200 nm (Zhang *et al.,* 2020). The insulin crystal is based on entry 1trz from the RCSB Protein Data Bank (https://www.rcsb.org). This crystal form has insulin hexamers packed into a rhombohedral H3 lattice, the most common space group observed for hexameric insulin structure determinations. A value of 1.22 g cm$^{-3}$ for proteins was used (Andersson and Hovmöller, 1998), with the asymmetric unit molecular weight obtained from the RCSB PDB structure summary page, yielding a volume occupancy of 0.686. The second most prevalent space group is a P2$_1$ lattice as exemplified with PDB ID 1ev6, which yields a volume occupancy of 0.584. Manual exceptions were made for several proteins: IAPP abundance was assigned 1% of insulin abundance (Caillon *et al.,* 2016); the copy number of vATPase was calculated by averaging copy numbers of the subunits with appropriate stoichiometry and the experimental copy number of carboxypeptidase E from Schvartz *et al.* (2012) was omitted from the copy number calculation because it is ~100 times greater than the other two proteomes.

An initial set of structures was gathered automatically from online databases using csvcomplete10.0.9 (https://github.com/brett barbaro/csvcomplete/blob/master/csvcomplete10.0.9.py). The UniProt RESTful API was used to retrieve sequences, and N-terminal signal sequences annotated in UniProt were automatically removed. The resulting sequences were then used to query the RCSB PDB BLAST-based RESTful API and return the top 10 structure matches in the PDB database. Subsequent manual curation was required in

several cases. Multiple files were needed to model cytoplasmic and lumen domains of Syt5, Pam, Atp6ap and Epha, and transmembrane segments were taken from integrin (PDB ID 2k1a). Similarly, files representing cytoplasmic and lumen domains of Ptprn and Ptprn2 were combined with transmembrane segments from receptor tyrosine kinase ErB1 (PDB ID 2m0b). All of the Vamps were modelled with Vamp-2, as PDB ID 2kog includes the entire protein in a lipid-bound environment. Granins were treated as coiled coils based on biophysical studies of chromogranin-A (Mosley *et al.,* 2007). Information was difficult to find for several proteins, resulting in poor or fragmentary structures for Nucb1, Nucb2, Dnajc2, Stc1 and Nptx1.

As we were completing this project and drafting this manuscript, structures generated by AlphaFold2 (Jumper *et al.,* 2021) became readily available through UniProt. We reevaluated the entire structural proteome to determine if these predicted structures added value. In the most challenging cases, such as the granins, AlphaFold2 provided structures that are largely unfolded and of low confidence. Predicted structures of Nucb1, Nucb2, Dnajc2, Stc1 and Nptx1 all showed a well-folded core similar to the existing homologous structures found by the methods above, flanked by long disordered regions of low confidence. These low-confidence regions could reflect intrinsic disorder in these structures or deficiencies in the AlphaFold2 method due to, for example, unmodelled oligomerisation. Ultimately, we chose to use one structure from AlphaFold2, for CD36, which showed a well-folded domain and two membrane-spanning helices, all consistent with UniProt annotations and existing partial structures. We envision that AlphaFold2 will become a more integral part of the structural pipeline in future, as methods are developed to create credible models that incorporate oligomerisation, definition of membrane-spanning regions and intrinsic disorder.

The cytoplasm was modelled using 50 cytoplasmic proteins with highest abundance in a recent proteomic study (Beck *et al.,* 2011; Supplementary Table 3). Relative abundances of these proteins were normalised to a total concentration of 0.2 g ml$^{-1}$ (Luby-Phelps, 1999).

### Model generation

Models were generated using a modified version of our instant distribution software (Klein *et al.,* 2018), based on cellPACK (Johnson *et al.,* 2015), similarly to the method used for building full models of *Mycoplasma genitalium* cells (Maritan *et al.,* 2022). During the packing process, a reduced representation of each protein is used, comprised of a list of representative beads or spheres selected using K-means clustering manually tuned to give a good ratio of number of beads to coverage of the entire protein. This manual tuning is particularly important for proteins with high aspect ratios, and is facilitated by the use of the web-based curation tool Mesoscope (Autin *et al.,* 2020). The beads are used to (i) estimate the space occupied by each protein in a master grid and (ii) relax overlapping proteins.

Molecules are distributed by a parallel algorithm that partitions the available space on a grid, places molecules, and performs a local relaxation to resolve conflicts. However, the insulin granule can consist of up to 2 million beads, so local relaxation could not be applied due to limitations in the NVIDIA Flex library to ~1 million beads. Thus, we exported and relaxed the model using LAMMPS (Thompson *et al.,* 2022) through Langevin rigid body dynamics. A custom approach based on cellPAINT (Gardner *et al.,* 2018, 2021) is used for the membrane proteins. Each membrane protein is augmented with two triads of beads on either side of the membrane

that constrain the protein within the membrane during the simulation. All bead interactions used a soft potential (https://docs.lammps.org/pair_soft.html).

The ultrastructure of the vesicle and boundary of the insulin crystal are defined as primitive signed distance fields (e.g. soft Boolean operation of spheres) or computed signed distance fields from a user-defined polygonal mesh (e.g. obtained from segmentation). In the current work, we used a spherical boundary for the crystal, and insulin was placed procedurally using the rhombohedral H3 lattice of PDB ID 1trz.

The instant packing method also includes a method for distributing lipids, using an approach similar to LipidWrapper (Durrant and Amaro, 2014). The vesicle membrane is represented by a spherical signed distance field, polygonised with a dual contour algorithm. The resulting triangulated surface is then tiled using small patches of lipids cookie-cut from equilibrated flat lipid bilayers. The operation is run in parallel on the gpu for each triangle giving interactive performance.

### Segmentation of X-ray tomograms

X-ray tomograms were obtained from the PBCC website (https://pbcconsortium.isrd.isi.edu). These datasets have a voxel width of 37.42 nm. Segmentation files were also obtained from the PBCC that define the location of the cell boundary, nucleus, endoplasmic reticulum and ISG locations. Blob detection was performed with VISFD (https://doi.org/10.5281/zenodo.5559243) in several steps. First, the position and orientation of the glass tube were identified automatically, and a mask was created to omit areas within 7–10 voxels of the tube from blob detection and visualisation. A mask was created to identify cytoplasmic regions inside the cell, using the PBC manual segmentation of the cell membrane and nucleus. Additional manual curation removed small features that were disconnected from the bulk of the cell. Blob detection was then performed on the maps, and blobs that overlap and blobs with weak scores were discarded. The threshold for discarding weak blobs was obtained by tuning the threshold for each map individually, choosing threshold values that correctly classify a test set of 30 manually chosen positives and 30 manually chosen negative decoys. Finally, an analysis of blobs was performed, including a radial density profile centred on the brightest pixel in the blob, distance from the cell surface and classification by brightness.

### Simulation of absorption of X-rays

From Ekman *et al.* (2018), image formation in soft X-ray tomography can be approximated as

$$-\ln \frac{I_{\mathrm{im}}}{I^0_{\mathrm{im}}} \approx \mathbf{A}_h \boldsymbol{\mu},$$

where $\mathbf{A}_h$ is a linear projection matrix incorporating the 3D point spread function of the system and $\boldsymbol{\mu}$ is a discretised vector representation of the linear attenuation coefficient (LAC) that is calculated from the atomistic model of the vesicle. Values of the mass attenuation coefficient (MAC; $\boldsymbol{\mu}/\rho$ where $\rho$ is the density) at a given photon energy (here 517 eV) are typically available for different materials (Henke *et al.*, 1993; http://henke.lbl.gov/optical_constants/atten2.html), and may be combined using the mixture rule, which is a mass-fraction-weighted average of the MAC of each component:

$$(\boldsymbol{\mu}/\rho)_{\mathrm{voxel}} = W_{\mathrm{protein}}(\boldsymbol{\mu}/\rho)_{\mathrm{protein}} + W_{\mathrm{lipid}}(\boldsymbol{\mu}/\rho)_{\mathrm{lipid}} + W_{\mathrm{water}}(\boldsymbol{\mu}/\rho)_{\mathrm{water}},$$

where $W_{\mathrm{protein}}$, $W_{\mathrm{lipid}}$ and $W_{\mathrm{water}}$ are the mass fraction of the component materials. The voxel protein LAC $\boldsymbol{\mu}_{\mathrm{voxel}}$ is then calculated as the MAC in the voxel $(\boldsymbol{\mu}/\rho)_{\mathrm{voxel}}$ times the protein density in the voxel $\rho_{\mathrm{voxel}}$ (protein weight/voxel volume):

$$\boldsymbol{\mu}_{\mathrm{voxel}} = (\boldsymbol{\mu}/\rho)_{\mathrm{voxel}} \times \rho_{\mathrm{voxel}}.$$

MAC values were calculated explicitly using the atomic composition of proteins in each voxel of the model, again using the mixture rule and the weight fraction of MAC values for each of the atom types. Lipids were based on DOPC ($C_{40}H_{80}NO_8P$), with a MAC value of 9,264.835 $\mathrm{cm}^{-1}$ and a volume of 1,150.0 $\mathrm{Å}^3$ per lipid (Greenwood *et al.*, 2006), and a value of 1,114.279 $\mathrm{cm}^{-1}$ was used for water.

To generate the final images, the initial calculated volume is embedded in a larger volume the size of the experimental data. The simulated reconstructions were obtained similar to the experimental ones: the projection images were distorted by Poisson noise corresponding to the shot noise of the experiment, and random translations were added to the tomograms to mimic subpixel alignment errors of the image registration (Chen *et al.*, 2022). The reconstruction was repeated 10 times for each volume and averaged for calculation of the radial profiles.

## Results and discussion

### Combining proteomic and annotation information

We manually curated a proteome based on the current state of knowledge for the system. This proteome is based largely on a comprehensive review (Suckale and Solimena, 2010), followed by literature searching to find supporting reports on each molecule. This includes 29 familiar proteins, including two forms of insulin, granins, enzymes involved in maturation, regulatory molecules and a variety of membrane-bound transporters and fusion-related proteins. These 29 proteins are included in Supplementary Table 1 with citations for the studies localising them to the ISG.

Manual curation, however, is intrinsically limited by the vagaries of literature search methods and user bias, so we developed a method to identify bona fide ISG proteins from proteomics data, removing false positives from ISG isolation impurities. Three proteomes were used in the current study, with a total of 270 proteins, of which 8 are observed in all three (two forms of insulin; converting enzymes Pcsk2 and Cpe; granins ChgA, ChgB and Scg2; nucleobindin-2). The studies used different protocols for isolation of ISG: proteome 1 used a gradient (Brunner *et al.*, 2007), proteome 2 added an affinity purification step (Hickey *et al.*, 2009) and proteome 3 used a three-step gradient purification along with stable isotope labelling with amino acids in cell culture (Schvartz *et al.*, 2012). Proteome 3 identified 668 proteins, but we included the 140 that were considered 'specific to ISGs', as done in a meta-analysis of all three studies (Crèvecoeur *et al.*, 2015).

The simplest approach to reconcile these differences would be to use the number of proteomic observations as a weighting factor. However, this has potential problems. For example, the converting factor Pcsk1, a necessary component of the ISG, showed up in only one study, but several subunits of mitochondrial ATP synthase, which is most likely a mitochondrial impurity, showed up in two. We developed an ROC-type scoring method that combines the number of proteomic observations and expert annotations of the cellular location. The study starts with the full set of proteins from

the three proteomes, and assigns our manually curated set as true positives and the rest as trial negatives. We then evaluate each location annotation for its ability to distinguish between the two. Annotations like 'cytoplasmic vesicle' have the power to discriminate true positives, and 'mitochondrion' and 'nucleus' are found for trial negatives (see Supplementary Table 2). A simple scoring function was developed to quantify this discrimination. Finally, an overall confidence score was developed that combines these location annotation scores with the number of times each protein is observed in a proteome, as described in the Methods section.

A manual literature search was then performed for all proteins with confidence >0.333, as well as all proteins present in two or more proteomes. As described below, in some cases, evidence for presence in the ISG was found, in other cases, evidence was found for presence in other compartments of the cell, and a few were ambiguous.

VAMPs are necessary for the fusion of vesicles with the membrane. Vamp2, Vamp3 and Vamp8 were included in our curated set. Vamp2 and Vamp3 scored well, but Vamp8, which has been immunolocalised to the ISG (Zhu et al., 2012), showed a midrange score of 0.21, due to being found in only one proteome and having lysosome and endosome localisations. Lamp2 also showed an intermediate score (0.76), but we could find no evidence for its presence in the ISG.

Rabs and similar membrane-associated proteins play an essential role in the regulation of the vesicle life cycle. Rab37, Rap1a and Rab3a were all included in the curated list, and showed high to intermediate scores. Immunolocalisation studies were found for Rph3al (Matsunaga et al., 2017), Rab3c and Rab3d (Iezzi et al., 1999). Rab27a is connected to secretory vesicles in general (Suckale and Solimena, 2010), but no evidence was found for several Rab5 members, Rab35 or Rhog. Rab1a, Rab2a and several G-protein subunits were found in two proteomes but received low scores because of localisation, and we could not find evidence for their presence in the ISG. Two additional proteins, Rac1 and Cdc42, were included based on the literature search (Wang and Thurmond, 2009), although they were not included in any of the proteomes.

Several proteins were not included in the original curated list, but strong evidence for their inclusion was found in the subsequent literature search. Peptide-amidating enzyme PAM (Garmendia et al., 2002), neurosecretory protein Vgf (Stephens, 2017), neuronalpentraxin-1 (Schvartz et al., 2012) and stanniocalcin-1 (Zaidi et al., 2012) all received high to intermediate scores. Conversely, several proteins were included in the original list, but scored poorly. Two nucleobindins, which have been placed in the ISG by immunolocalisation (Ramesh et al., 2015), scored low due to nuclear and endoplasmic reticulum annotations, in spite of nucleobindin-2 being found in all three proteomic studies. vATPase has been localised to the ISG (Sun-Wada et al., 2006), but most of the subunits scored poorly, due to their presence in a single proteome and a variety of conflicting localisations. Macrophage migration inhibitory factor was found in only one proteome, but has been immunolocalised to the ISG (Waeber et al., 1997).

Several proteins showed high scores, and evocative but nonconclusive evidence was found in the literature search. Tmem163, a putative zinc transporter (Sharma et al., 2017), and Enpp2 (Gorelik et al., 2017) have connections to type 2 diabetes. Depalmatase Abhd17 (Won et al., 2018), Dnajc5 (Gorenberg and Chandra, 2017) and Wnt ligands like Wif1 are involved in the secretory process (Schinner et al., 2007). These may be candidates for immunolocalisation studies.

A variety of proteins were found in two of the proteomic studies, but appear to be impurities from other compartments in the cells. These include lysosomal proteins alpha-glucosidase, septin-11 and cathepsin D; cytoplasmic enzymes ATP-citrate synthase, fructose bisphosphate aldolase A and glyceraldehyde-3-phosphate dehydrogenase; a subunit of mitochondrial ATP synthase and tubulin and fibronectin. All showed low scores due to the localisation annotations and were culled from the list used for modelling.

## Models of the insulin secretory granule

Idealised models of mature and immature vesicles were created with the final proteome of 56 proteins (Supplementary Table 1), an insulin crystal with a diameter of 200 nm (Zhang et al., 2020), and a vesicle diameter of 320 nm (Suckale and Solimena, 2010). The initial model created with the interactive method of cellPACKgpu roughly places molecules in the proper compartments, but has a number of steric contacts, particularly with highly asymmetric molecules such as the granins. Subsequent optimisation with LAMMPS resolves these contacts to create the final model. Relaxation is required between 1 and 2 h using 30 threads, followed by ~30 min to recompute the position/rotation of the protein instance from the bead coordinates for each frame of the simulation (CPU AMD Ryzen Threadripper 1950X 16-Core Processor, GPU NVIDIA Quadro R8000). The final models reveal the ISG as being densely packed with soluble proteins and with a protein-rich bounding membrane. The full model (idealised mature and immature granules surrounded by cytoplasmic proteins) consists of 424,384 individual proteins represented by 2,343,538 beads of radius 17.0 Å.

## Interpreting experimental soft X-ray tomograms

Fig. 2 demonstrates the use of this methodology to interpret experimental soft X-ray tomograms. Tomograms of whole pancreatic beta cells show many easily distinguishable features with high LAC values. The ones with the highest LAC values are presumed to be lipid droplets, and the remainder are insulin secretory vesicles. We address two questions that have been posed with the experimental work.

First, there has been some question about the contribution of the membrane to the observed features. We created models and simulated tomograms for vesicles with and without the membrane (Fig. 2) and calculated a 2D radial profile for all four volumes. These show that the membrane for these idealised vesicles accounts for 10–20% of the observed X-ray absorbance in these features. For the full model with lipids, the projected mature blob peak value is 0.538 $\mu m^{-1}$ for mature and 0.437 $\mu m^{-1}$ for immature, and the surrounding cytoplasm has an average value of 0.35 $\mu m^{-1}$.

Second, it has been noted that immature vesicles, which do not include the dense crystal of insulin, may not show enough contrast to be observed in experimental soft X-ray tomograms. As seen in Fig. 2, immature vesicles are less visible in simulated tomograms. When we applied our VISFD segmentation protocol to these simulated tomograms, only the mature vesicles were detected with a contour diameter of 184 nm, whereas the immature vesicles were undetected. This result indicates that a pool of immature vesicles may not be detected with the VISFD segmentation protocol in the analysis described below. We also evaluated the need for the relaxation step in model generation, when these models are used to simulate soft X-ray tomograms, and as seen in the graph in Fig. 2,

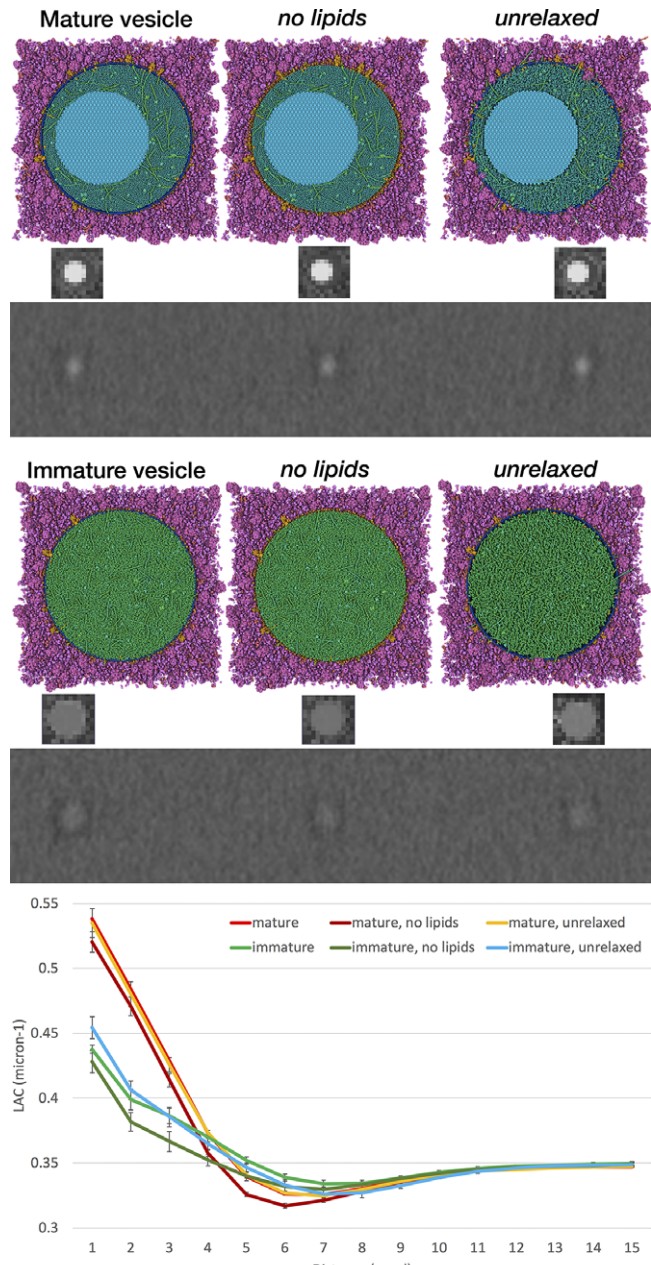

**Figure 2.** Simulation of X-ray tomograms. Mature and immature vesicles are shown in three states: the full model, a model without lipids and a rough model generated without the relaxation step. The small insets are 'phantoms': voxelised representations of the simulated linear attenuation coefficient for each vesicle. These phantoms are embedded in a large volume of cytoplasm and used to simulate tomograms that reflect the experimental imaging and processing, as shown below the models and phantoms. Radial density profiles for the six models are shown at the bottom.

unrelaxed and relaxed models show very similar radial profiles. In the following larger study, a quick optimisation was performed to ensure that all lumen proteins are inside the membrane, but a full relaxation was not performed.

We then estimated the size and insulin content of all visible vesicles in an experimental whole-cell tomogram (cell 766_5; Fig. 3*a*). We segmented the experimental volume with VISFD, resulting in 1,136 features interpreted to be vesicles (Fig. 3*e*).

We generated a library of 3,456 models to sample potential variations in vesicle and cytoplasmic properties (4 h of computation). Sixteen vesicle diameters were sampled from 130 to 473 nm. For each vesicle size, six different insulin crystal sizes were sampled and the number of soluble and membrane proteins was scaled based on the number of insulin molecules in the crystal. Six steps of transition between mature and immature vesicles were generated by partitioning this amount of insulin as mature insulin in a crystal, proinsulin in immature vesicles, and cleaved insulin monomer in the lumen in transitional forms. Six different concentrations of cytosolic proteins were also scanned from 0.06 to 0.2 g ml$^{-1}$. A quick relaxation of 150 iterations was applied on the gpu to force proteins (in particular, the highly extended granins) inside the vesicles. Finally, model LAC values were computed for each of these sample vesicles in a 636-nm bounding box (17 × 17 × 17 voxels with a voxel size of 37.42 nm), then embedded in a surrounding volume of size 247 × 17 × 247 voxels (Fig. 3*b,c*). Each volume was projected and reconstructed 10 times (37 h of computation), used to calculate an average 2D radial profile, and an $R^2$ score between the experimental profile and the simulated profile was calculated, yielding a 'confidence level' in the assignment. $R^2$ scores are calculated for six radial distance points to focus on the vesicle profile, and for 15 points to include information on the surrounding cytoplasmic LAC. The simulated profile with the maximum $R^2$ score was selected for each of the 1,136 features in the experimental tomogram.

We obtained an average $R^2$ score of 0.94 ± 0.1 across all 1,136 experimental blobs, with 83% with a score >0.93 and 68% with a score >0.96. Looking at the steps of transition, 128 blobs are assigned as immature vesicles, 747 in transitional states and 261 assigned as mature vesicles (Fig. 3*f*). This predominance of immature and transitional forms is consistent with recent cryoelectron tomography studies, which found that the percentage of mature vesicles ranged from 12 to 32% depending on the location in the cell (Zhang *et al.*, 2020). In addition, note that this analysis may be missing a pool of immature vesicles that are not visible in the soft X-ray tomographic experiment, as described above. Several examples are shown in detail in Fig. 4.

## Outlook

The convergence of experimental methodology and computational capability is putting whole-cell structural modelling within reach. Using this methodology, it is becoming possible to interpret entire cellular tomograms with integrative molecular models. In Fig. 3, an entire X-ray tomogram has been segmented to identify mature secretory vesicles, and models have been generated with crystal sizes that match the observed absorption of the vesicle. This model is the first step towards a quantitative mesoscale interpretation of this cell. For example, in cell 766_5 from the PBCC, the observed features include a total of ~606,000,000 insulin monomers across the cell, with an average of 533,000 ± 253,000 per vesicle, and with ~125,000,000 out of the total in crystal form.

One of the goals of our work on the cellPACK suite is to create tools that will be widely usable; however, these models are straining the current capabilities of consumer-level computational hardware and software. The cellPACK suite heavily leverages advances from the gaming community for use of gpu hardware, which underlies the instant packing algorithm used to generate initial models (Klein *et al.*, 2018). For the relaxation/optimisation steps, the size of these models required a move from the interactive

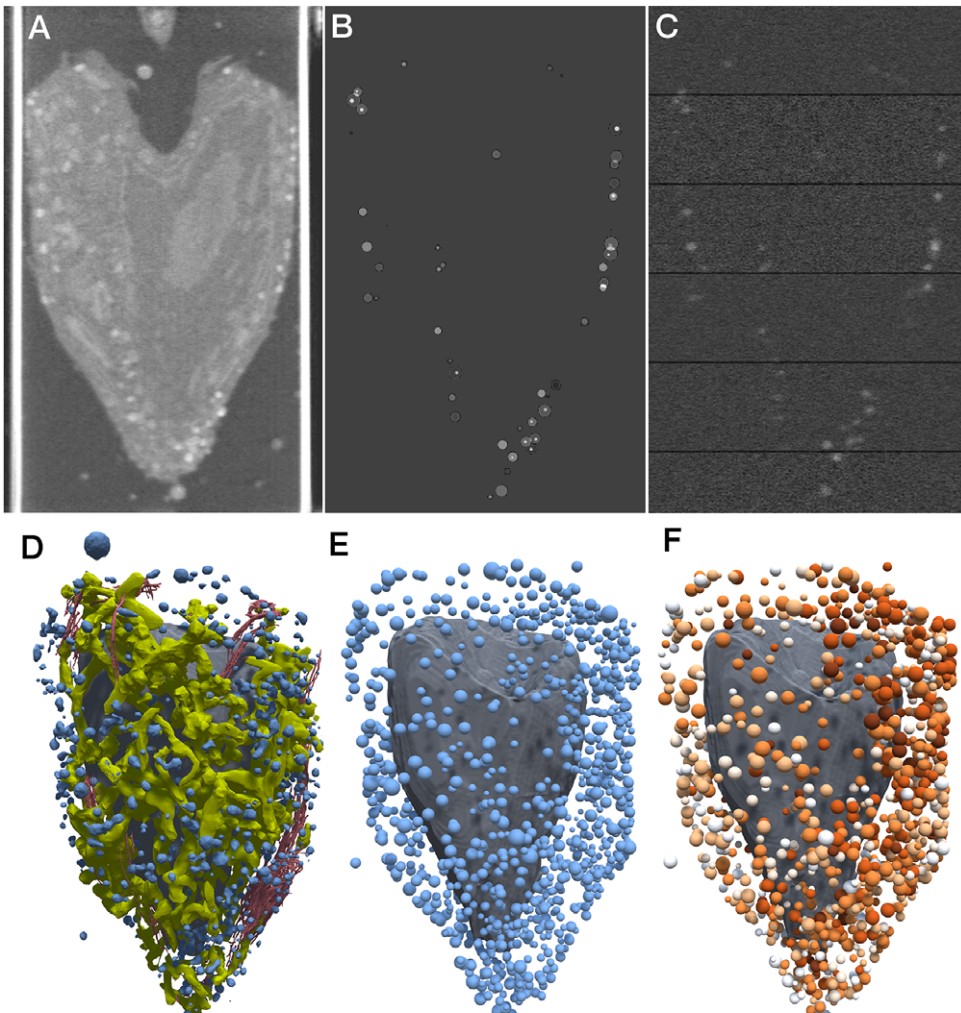

**Figure 3.** Interpretation of an experimental X-ray tomogram. (*a*) Volume rendering of a slice through the X-ray tomogram of cell 766_5 from the Pancreatic Beta-Cell Consortium (PBCC). Bright white vertical bands at the edges are the capillary used to hold the cell. (*b*) Idealised simulated absorption for this slice from vesicle models placed at features in the tomogram. (*c*) Simulated absorption of this slice mimicking the experimental imaging and processing. Horizontal bands are due to the calculation of the volume in sections. (*d*) Manual segmentation from the PBCC showing mitochondria (yellow), nucleus (grey), endoplasmic reticulum (red) and vesicles (blue). (*e*) VISFD automatically segments blob-like features as spheres (blue). The manually segmented nucleus is included in grey for context. (*f*) Interpretation of the automated segmentation with idealised spherical vesicles, showing the predicted vesicle membrane radius and coloured with mature vesicles in dark red, immature vesicles in white and transitional forms in pastel shades.

methods of Flex to the traditional batch mode of LAMMPS. We thus developed an initial simplistic input for LAMMPS that will serve as the foundation for more advanced simulation of similar large systems. Finally, visualisation before and after the simulation is provided by cellVIEW (Muzic *et al.,* 2015) within our cellPACK application built within Unity. Simulation can be cached and played back directly in our application. Final models are saved/ exported in different resolutions and file formats (.bin, .pdb and .cif) to enable visualisation with other molecular graphics software (e.g. OVITO, VMD, UCSF Chimera and Mol*). These files typically include coordinates for each type of protein along with transformation information for placing all of the instances of the protein into the overall model. However, as of today, only Mol* (Sehnal *et al.,* 2018) is capable of reading and visualising the full all-atom model (Fig. 5) and this at a low frame rate using a coarse Gaussian surface representation of the molecules (e.g. visit

the GitHub site included below to see Mol* visualisation of the mature and immature vesicles). Other molecular graphics tools were able to load coordinate files for each type of protein but currently failed to build the many instances of each protein to visualise the entire model.

Of course, many challenges remain as we move forward in this mesoscale structural view of a cell. As is always the case with biology, there are numerous unique features for each of the compartments of the cell, and new methodologies will be needed, for example, to incorporate complex structural features like intrinsic disorder in nuclear pores, the cytoskeleton and its interactions with organelles, the dynamic endomembrane system and the proteins that manage it and the many structural and functional states of chromatin. With all of this, structural mesoscale modelling will proceed with a combination of automated methods when possible, and manual attention when necessary.

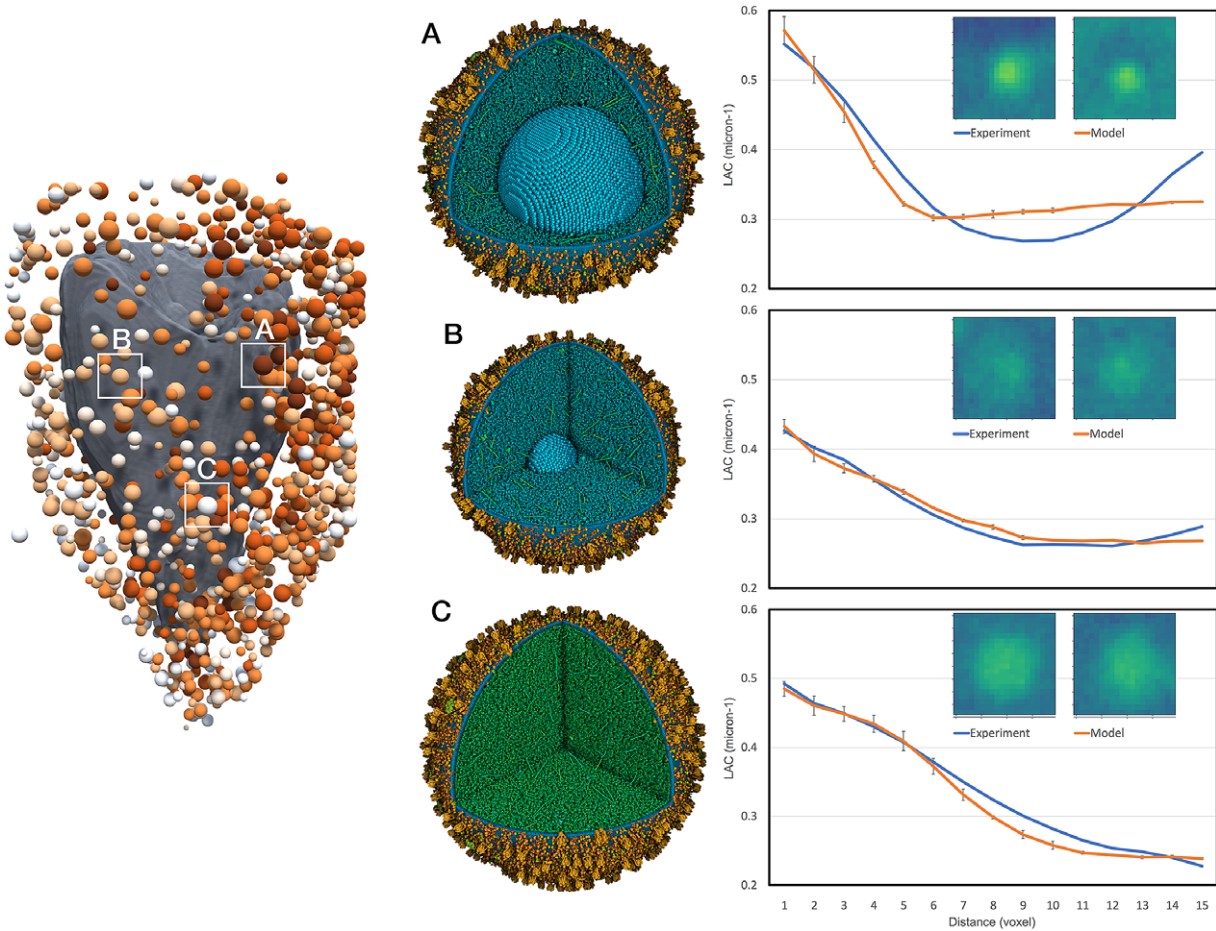

**Figure 4.** Interpretation of selected features in the experimental tomogram. Three vesicles are shown: (*a*) mature vesicle with a radius of 236 nm and a crystal of 124 nm, and cytoplasmic concentration to give an average linear attenuation coefficient (LAC) of 0.34 $\mu m^{-1}$, (*b*) transitional vesicle with a radius of 202 nm and a crystal of 50 nm, and cytoplasmic LAC of 0.28 $\mu m^{-1}$ and (*c*) immature vesicle with a radius of 236 nm and cytoplasmic LAC of 0.25 $\mu m^{-1}$. The vesicle models are shown at the centre and experimental (blue curve) and simulated (orange curve) radial density profiles are shown at the right with a central slice through the tomograms.

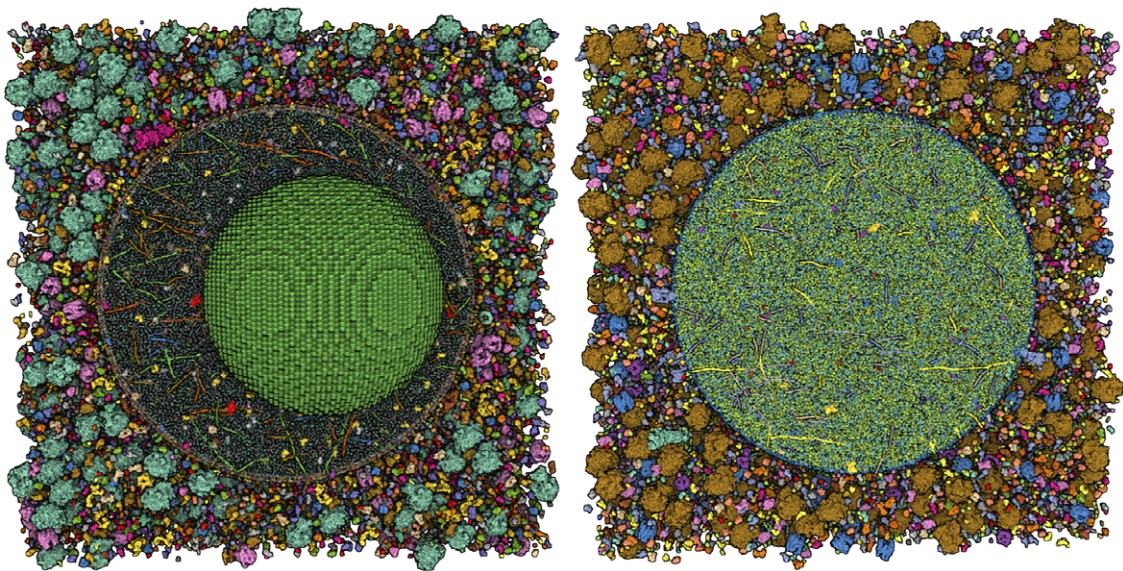

**Figure 5.** Idealised models of (left) mature and (right) immature vesicles viewed interactively in Mol* using a coarse Gaussian surface and coloured by the default "Color by Chain Id" property.

**Acknowledgements.** We thank Kate L. White, Xianjun Zhang, Carolyn Larabell, Yu Gao, and Raymond C. Stevens for helpful discussions.

**Supplementary Materials.** To view supplementary material for this article, please visit http://doi.org/10.1017/qrd.2022.10.

**Data availability statement.** Models and volumes for the mature and immature vesicles are available at https://github.com/ccsb-scripps/ISG. Links to Mol* visualisations are also available for mature and immature models at the GitHub site.

**Author contributions** Investigation: L.A., B.A.B., A.I.J., S.V., A.J.O., D.S.G.; Methodology: L.A., B.A.B., A.I.J., A.E., S.V., D.S.G.; Writing: L.A., B.A.B., A.I.J., A.E., A.J.O., D.S.G.

**Financial support.** This work was supported by the NIH R01 GM120604 (DSG) and the Pancreatic Beta Cell Consortium (BAB). The National Center for X-Ray Tomography was supported by the NIH NIGMS (P30GM138441) and the DOE Office of Biological and Environmental Research (DE-AC02-5CH11231), and is located at the Advanced Light Source, a U.S. DOE Office of Science User Facility under contract no. DE-AC02-05CH11231. RCSB PDB (DSG) is funded by the National Science Foundation (DBI-1832184), the U.S. Department of Energy (DE-SC0019749), and the National Cancer Institute, National Institute of Allergy and Infectious Diseases, and National Institute of General Medical Sciences of the National Institutes of Health (R01GM133198).

**Conflict of interest.** The authors declare no conflicts of interest.

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
