## [Reviewer Report]

*Comments to Author*: The paper addresses an important problem of generating in silico models of cellular environment at atomic resolution. Such models can be used as unique tools to study cellular biology, to complement rather sparse experimental data on high-resolution cellular organization. Importantly, they can be used as starting points for dynamic simulation of molecular mechanism inside a cell. The authors are well-recognized leaders in mesoscale structural modeling, with a long and distinguished record of accomplishments in this strategic area of life science. Their integrative modeling approach learns structural parameters from experimental observations, arranges the constituent components according to estimated density and localization, and compares the resulting arrangements with experimental observations, such as X-ray tomograms at the molecular level. The approach is rigorous and well documented, the data is publicly available, and the paper is well written and illustrated. As such, it should be of great interest to the research community.

---

## [Reviewer Report]

*Comments to Author*: In the present manuscript the authors are part of Pancreatic Beta-Cell Consortium (PBCC) and they are developing methods to generate mesoscale models of functional regions of the pancreatic beta cell based on diverse experimental data. In the manuscript the authors have chosen to model the insulin secretory granule (ISG). They have presented an entire pipeline from data curation to model generation, and present potential applications. They also have shown preliminary results of soft X-ray tomograms simulations of vesicles, exploring the use of integrative modeling to provide molecule-level interpretation of these tomograms.

I believe this work represents an enormous step-forward in the structural mesoscale modeling and visualization of a cellular organelle, and will be of enormous value to the simulation research and education communities. There is a strong movement in the computational chemistry field towards distributed approaches, particularly for non-developers. The materials described and distributed with this manuscript provide an exceptional example / set a new standard for making the leap from publishing code in github to having a tool that can be used by anyone.

I strongly endorse publication in QRB.

I choose to review non-anonymously,

Pablo Ricardo Arantes

---

## [Reviewer Report]

*Comments to Author*: Reviewer #1: The paper addresses an important problem of generating in silico models of cellular environment at atomic resolution. Such models can be used as unique tools to study cellular biology, to complement rather sparse experimental data on high-resolution cellular organization. Importantly, they can be used as starting points for dynamic simulation of molecular mechanism inside a cell. The authors are well-recognized leaders in mesoscale structural modeling, with a long and distinguished record of accomplishments in this strategic area of life science. Their integrative modeling approach learns structural parameters from experimental observations, arranges the constituent components according to estimated density and localization, and compares the resulting arrangements with experimental observations, such as X-ray tomograms at the molecular level. The approach is rigorous and well documented, the data is publicly available, and the paper is well written and illustrated. As such, it should be of great interest to the research community.

Reviewer #2: In the present manuscript the authors are part of Pancreatic Beta-Cell Consortium (PBCC) and they are developing methods to generate mesoscale models of functional regions of the pancreatic beta cell based on diverse experimental data. In the manuscript the authors have chosen to model the insulin secretory granule (ISG). They have presented an entire pipeline from data curation to model generation, and present potential applications. They also have shown preliminary results of soft X-ray tomograms simulations of vesicles, exploring the use of integrative modeling to provide molecule-level interpretation of these tomograms.

I believe this work represents an enormous step-forward in the structural mesoscale modeling and visualization of a cellular organelle, and will be of enormous value to the simulation research and education communities. There is a strong movement in the computational chemistry field towards distributed approaches, particularly for non-developers. The materials described and distributed with this manuscript provide an exceptional example / set a new standard for making the leap from publishing code in github to having a tool that can be used by anyone.

I strongly endorse publication in QRB.

I choose to review non-anonymously,

Pablo Ricardo Arantes